# The Neural Hawkes Process: A Neurally Self-Modulating Multivariate Point Process

**Hongyuan Mei**     **Jason Eisner**
Department of Computer Science, Johns Hopkins University
3400 N. Charles Street, Baltimore, MD 21218 U.S.A
{hmei,jason}@cs.jhu.edu

## Abstract

Many events occur in the world. Some event types are stochastically excited or inhibited—in the sense of having their probabilities elevated or decreased—by patterns in the sequence of previous events. Discovering such patterns can help us predict *which type* of event will happen next and *when*. We model streams of discrete events in continuous time, by constructing a **neurally self-modulating multivariate point process** in which the intensities of multiple event types evolve according to a novel **continuous-time LSTM**. This generative model allows past events to influence the future in complex and realistic ways, by conditioning future event intensities on the hidden state of a recurrent neural network that has consumed the stream of past events. Our model has desirable qualitative properties. It achieves competitive likelihood and predictive accuracy on real and synthetic datasets, including under missing-data conditions.

## 1   Introduction

Some events in the world are correlated. A single event, or a pattern of events, may help to cause or prevent future events. We are interested in learning the distribution of sequences of events (and in future work, the causal structure of these sequences). The ability to discover correlations among events is crucial to accurately predict the future of a sequence given its past, i.e., which events are likely to happen next and when they will happen.

We specifically focus on sequences of discrete events in continuous time ("event streams"). Modeling such sequences seems natural and useful in many applied domains:

- *Medical events.* Each patient has a sequence of acute incidents, doctor's visits, tests, diagnoses, and medications. By learning from previous patients what sequences tend to look like, we could predict a new patient's future from their past.

- *Consumer behavior.* Each online consumer has a sequence of online interactions. By modeling the distribution of sequences, we can learn purchasing patterns. Buying cookies may temporarily depress purchases of all desserts, yet increase the probability of buying milk.

- *"Quantified self" data.* Some individuals use cellphone apps to record their behaviors—eating, traveling, working, sleeping, waking. By anticipating behaviors, an app could perform helpful supportive actions, including issuing reminders and placing advance orders.

- *Social media actions.* Previous posts, shares, comments, messages, and likes by a set of users are predictive of their future actions.

- Other event streams arise in *news*, *animal behavior*, *dialogue*, *music*, etc.

A basic model for event streams is the **Poisson process** (Palm, 1943), which assumes that events occur independently of one another. In a **non-homogenous Poisson process**, the (infinitesimal) probability of an event happening at time $t$ may vary with $t$, but it is still independent of other events. A **Hawkes process** (Hawkes, 1971; Liniger, 2009) supposes that past events can temporarily *raise* the probability of future events, assuming that such excitation is ① positive, ② additive over the past events, and ③ exponentially decaying with time.

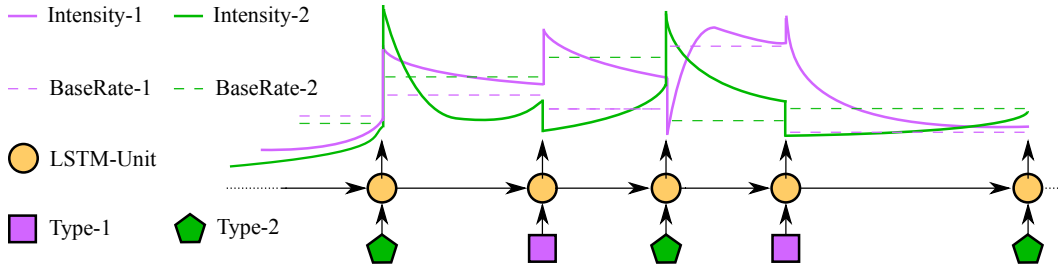

Figure 1: Drawing an event stream from a neural Hawkes process. An LSTM reads the sequence of past events (polygons) to arrive at a hidden state (orange). That state determines the future "intensities" of the two types of events—that is, their time-varying instantaneous probabilities. The intensity functions are continuous parametric curves (solid lines) determined by the most recent LSTM state, with dashed lines showing the steady-state asymptotes that they would eventually approach. In this example, events of type 1 excite type 1 but inhibit type 2. Type 2 excites itself, and excites or inhibits type 1 according to whether the count of type 2 events so far is odd or even. Those are immediate effects, shown by the sudden jumps in intensity. The events also have longer-timescale effects, shown by the shifts in the asymptotic dashed lines.

However, real-world patterns often seem to violate these assumptions. For example, ① is violated if one event inhibits another rather than exciting it: cookie consumption inhibits cake consumption. ② is violated when the combined effect of past events is not additive. Examples abound: The 20th advertisement does not increase purchase rate as much as the first advertisement did, and may even drive customers away. Market players may act based on their own complex analysis of market history. Musical note sequences follow some intricate language model that considers melodic trajectory, rhythm, chord progressions, repetition, etc. ③ is violated when, for example, a past event has a delayed effect, so that the effect starts at 0 and increases sharply before decaying.

We generalize the Hawkes process by determining the event intensities (instantaneous probabilities) from the hidden state of a recurrent neural network. This state is a deterministic function of the past history. It plays the same role as the state of a deterministic finite-state automaton. However, the recurrent network enjoys a continuous and infinite state space (a high-dimensional Euclidean space), as well as a learned transition function. In our network design, the state is updated discontinuously with each successive event occurrence and also evolves continuously as time elapses between events.

Our main motivation is that our model can capture effects that the Hawkes process misses. The combined effect of past events on future events can now be superadditive, subadditive, or even subtractive, and can depend on the sequential ordering of the past events. Recurrent neural networks already capture other kinds of complex sequential dependencies when applied to language modeling—that is, generative modeling of linguistic word sequences, which are governed by syntax, semantics, and habitual usage (Mikolov et al., 2010; Sundermeyer et al., 2012; Karpathy et al., 2015). We wish to extend their success (Chelba et al., 2013) to sequences of events in *continuous* time.

Another motivation for a more expressive model than the Hawkes process is to cope with missing data. Even in a domain where Hawkes might be appropriate, it is hard to apply Hawkes when sequences are only partially observed. Real datasets may *systematically* omit some types of events (e.g., illegal drug use, or offline purchases) which, in the true generative model, would have a strong influence on the future. They may also have *stochastically* missing data, where the missingness mechanism—the probability that an event is not recorded—can be complex and data-dependent (MNAR). In this setting, we can fit our model directly to the observation sequences, and use it to predict observation sequences that were generated in the same way (using the same complete-data distribution and the same missingness mechanism). Note that if one knew the true complete-data distribution—perhaps Hawkes—and the true missingness mechanism, one would optimally predict the incomplete future from the incomplete past in Bayesian fashion, by integrating over possible completions (imputing the missing events and considering their influence on the future). Our hope is that the neural model is expressive enough that it can learn to approximate this true predictive distribution. Its hidden state after observing the past should implicitly encode the Bayesian posterior, and its update rule for this hidden state should emulate the "observable operator" that updates the posterior upon each new observation. See Appendix A.4 for further discussion.

A final motivation is that one might wish to *intervene* in a medical, economic, or social event stream so as to improve the future course of events. Appendix D discusses our plans to deploy our model family as an environment model within reinforcement learning, where an agent controls some events.

## 2   Notation

We are interested in constructing distributions over **event streams** $(k_1, t_1), (k_2, t_2), \ldots$, where each $k_i \in \{1, 2, \ldots, K\}$ is an event type and $0 < t_1 < t_2 < \cdots$ are times of occurrence.[1] That is, there are $K$ types of events, tokens of which are observed to occur in continuous time.

For any distribution $P$ in our proposed family, an event stream is almost surely infinite. However, when we observe the process only during a time interval $[0, T]$, the number $I$ of observed events is almost surely finite. The *log-likelihood* $\ell$ of the model $P$ given these $I$ observations is

$$\Big( \sum_{i=1}^{I} \log P((k_i, t_i) \mid \mathcal{H}_i) \Big) + \log P(t_{I+1} > T \mid \mathcal{H}_I) \tag{1}$$

where the **history** $\mathcal{H}_i$ is the prefix sequence $(k_1, t_1), (k_2, t_2), \ldots, (k_{i-1}, t_{i-1})$, and $P((k_i, t_i) \mid \mathcal{H}_i)$ is the probability density that the *next* event occurs at time $t_i$ and has type $k_i$.

Throughout the paper, the subscript $i$ usually denotes quantities that affect the distribution of the next event $(k_i, t_i)$. These quantities depend only on the history $\mathcal{H}_i$.

We use (lowercase) Greek letters for parameters related to the classical Hawkes process, and Roman letters for other quantities, including hidden states and affine transformation parameters. We denote vectors by bold lowercase letters such as $\mathbf{s}$ and $\boldsymbol{\mu}$, and matrices by bold capital Roman letters such as $\mathbf{U}$. Subscripted bold letters denote distinct vectors or matrices (e.g., $\mathbf{w}_k$). Scalar quantities, including vector and matrix elements such as $s_k$ and $\alpha_{j,k}$, are written without bold. Capitalized scalars represent upper limits on lowercase scalars, e.g., $1 \le k \le K$. Function symbols are notated like their return type. All $\mathbb{R} \to \mathbb{R}$ functions are extended to apply elementwise to vectors and matrices.

## 3   The Model

In this section, we first review Hawkes processes, and then introduce our model one step at a time.

Formally, generative models of event streams are **multivariate point processes**. A (temporal) point process is a probability distribution over $\{0, 1\}$-valued functions on a given time interval (for us, $[0, \infty)$). A multivariate point process is formally a distribution over $K$-tuples of such functions. The $k^{\text{th}}$ function indicates the times at which events of type $k$ occurred, by taking value 1 at those times.

### 3.1   Hawkes Process: A Self-Exciting Multivariate Point Process (SE-MPP)

A basic model of event streams is the **non-homogeneous multivariate Poisson process**. It assumes that an event of type $k$ occurs at time $t$—more precisely, in the infinitesimally wide interval $[t, t + dt)$—with probability $\lambda_k(t)dt$. The value $\lambda_k(t) \ge 0$ can be regarded as a rate per unit time, just like the parameter $\lambda$ of an ordinary Poisson process. $\lambda_k$ is known as the **intensity function**, and the total intensity of all event types is given by $\lambda(t) = \sum_{k=1}^{K} \lambda_k(t)$.

A well-known generalization that captures interactions is the **self-exciting multivariate point process (SE-MPP)**, or **Hawkes process** (Hawkes, 1971; Liniger, 2009), in which past events $h$ from the history conspire to *raise* the intensity of each type of event. Such excitation is positive, additive over the past events, and exponentially decaying with time:

$$\lambda_k(t) = \mu_k + \sum_{h:t_h<t} \alpha_{k_h,k} \exp(-\delta_{k_h,k}(t - t_h)) \tag{2}$$

where $\mu_k \ge 0$ is the base intensity of event type $k$, $\alpha_{j,k} \ge 0$ is the degree to which an event of type $j$ initially excites type $k$, and $\delta_{j,k} > 0$ is the decay rate of that excitation. When an event occurs, all intensities are elevated to various degrees, but then will decay toward their base rates $\boldsymbol{\mu}$.

## 3.2 Self-Modulating Multivariate Point Processes

The positivity constraints in the Hawkes process limit its expressivity. First, the positive interaction parameters $\alpha_{j,k}$ fail to capture inhibition effects, in which past events *reduce* the intensity of future events. Second, the positive base rates $\boldsymbol{\mu}$ fail to capture the inherent inertia of some events, which are unlikely until their cumulative excitation by past events crosses some threshold. To remove such limitations, we introduce two *self-modulating* models. Here the intensities of future events are stochastically *modulated* by the past history, where the term "modulation" is meant to encompass both excitation and inhibition. The intensity $\lambda_k(t)$ can even fluctuate non-monotonically between successive events, because the competing excitatory and inhibitory influences may decay at different rates.

### 3.2.1 Hawkes Process with Inhibition: A Decomposable Self-Modulating MPP (D-SM-MPP)

Our first move is to enrich the Hawkes model's expressiveness while still maintaining its decomposable structure. We relax the positivity constraints on $\alpha_{j,k}$ and $\mu_k$, allowing them to range over $\mathbb{R}$, which allows *inhibition* ($\alpha_{j,k} < 0$) and *inertia* ($\mu_k < 0$). However, the resulting total activation could now be negative. We therefore pass it through a non-linear **transfer function** $f_k : \mathbb{R} \to \mathbb{R}_+$ to obtain a positive intensity function as required:

$$\lambda_k(t) = f_k(\tilde{\lambda}_k(t)) \quad \text{(3a)} \qquad \tilde{\lambda}_k(t) = \mu_k + \sum_{h:t_h<t} \alpha_{k_h,k} \exp(-\delta_{k_h,k}(t - t_h)) \qquad \text{(3b)}$$

As $t$ increases between events, the intensity $\lambda_k(t)$ may both rise and fall, but eventually approaches the base rate $f(\mu_k+0)$, as the influence of each previous event still decays toward 0 at a rate $\delta_{j,k} > 0$.

What non-linear function $f_k$ should we use? The ReLU function $f(x) = \max(x, 0)$ is not strictly positive as required. A better choice is the scaled "softplus" function $f(x) = s \log(1 + \exp(x/s))$, which approaches ReLU as $s \to 0$. We learn a separate scale parameter $s_k$ for each event type $k$, which adapts to the rate of that type. So we instantiate (3a) as $\lambda_k(t) = f_k(\tilde{\lambda}_k(t)) = s_k \log(1 + \exp(\tilde{\lambda}_k(t)/s_k))$. Appendix A.1 graphs this and motivates the "softness" and the scale parameter.

### 3.2.2 Neural Hawkes Process: A Neurally Self-Modulating MPP (N-SM-MPP)

Our second move removes the restriction that the past events have independent, additive influence on $\tilde{\lambda}_k(t)$. Rather than predict $\tilde{\lambda}_k(t)$ as a simple summation (3b), we now use a recurrent neural network. This allows learning a complex dependence of the intensities on the number, order, and timing of past events. We refer to our model as a **neural Hawkes process**.

Just as before, each event type $k$ has an time-varying intensity $\lambda_k(t)$, which jumps discontinuously at each new event, and then drifts continuously toward a baseline intensity. In the new process, however, these dynamics are controlled by a hidden state vector $\mathbf{h}(t) \in (-1, 1)^D$, which in turn depends on a vector $\mathbf{c}(t) \in \mathbb{R}^D$ of memory cells in a **continuous-time LSTM**.[2] This novel recurrent neural network architecture is inspired by the familiar discrete-time LSTM (Hochreiter and Schmidhuber, 1997; Graves, 2012). The difference is that in the continuous interval following an event, each memory cell $c$ *exponentially decays* at some rate $\delta$ toward some steady-state value $\bar{c}$.

At each time $t > 0$, we obtain the intensity $\lambda_k(t)$ by (4a), where (4b) defines how the hidden states $\mathbf{h}(t)$ are continually obtained from the memory cells $\mathbf{c}(t)$ as the cells decay:

$$\lambda_k(t) = f_k(\mathbf{w}_k^\top \mathbf{h}(t)) \qquad \text{(4a)} \qquad \mathbf{h}(t) = \mathbf{o}_i \odot (2\sigma(2\mathbf{c}(t)) - 1) \text{ for } t \in (t_{i-1}, t_i] \qquad \text{(4b)}$$

This says that on the interval $(t_{i-1}, t_i]$—in other words, after event $i-1$ up until event $i$ occurs at some time $t_i$—the $\mathbf{h}(t)$ defined by equation (4b) determines the intensity functions via equation (4a). So for $t$ in this interval, according to the model, $\mathbf{h}(t)$ is a sufficient statistic of the history $(\mathcal{H}_i, t - t_{i-1})$ with respect to future events (see equation (1)). $\mathbf{h}(t)$ is analogous to $\mathbf{h}_i$ in an LSTM language model (Mikolov et al., 2010), which summarizes the past event sequence $k_1, \dots, k_{i-1}$. But in our decay architecture, it will also reflect the interarrival times $t_1 - 0, t_2 - t_1, \dots, t_{i-1} - t_{i-2}, t - t_{i-1}$.

This interval $(t_{i-1}, t_i]$ ends when the next event $k_i$ stochastically occurs at some time $t_i$. At this point, the continuous-time LSTM reads $(k_i, t_i)$ and updates the current (decayed) hidden cells $\mathbf{c}(t)$ to new initial values $\mathbf{c}_{i+1}$, based on the current (decayed) hidden state $\mathbf{h}(t_i)$.

How does the continuous-time LSTM make those updates? Other than depending on decayed values $\mathbf{h}(t_i)$, the update formulas resemble the discrete-time case:[3]

$$\mathbf{i}_{i+1} \leftarrow \sigma\left(\mathbf{W}_\mathrm{i}\mathbf{k}_i + \mathbf{U}_\mathrm{i}\mathbf{h}(t_i) + \mathbf{d}_\mathrm{i}\right) \quad (5a)$$

$$\mathbf{f}_{i+1} \leftarrow \sigma\left(\mathbf{W}_\mathrm{f}\mathbf{k}_i + \mathbf{U}_\mathrm{f}\mathbf{h}(t_i) + \mathbf{d}_\mathrm{f}\right) \quad (5b)$$

$$\mathbf{z}_{i+1} \leftarrow 2\sigma\left(\mathbf{W}_\mathrm{z}\mathbf{k}_i + \mathbf{U}_\mathrm{z}\mathbf{h}(t_i) + \mathbf{d}_\mathrm{z}\right) - 1 \quad (5c)$$

$$\mathbf{o}_{i+1} \leftarrow \sigma\left(\mathbf{W}_\mathrm{o}\mathbf{k}_i + \mathbf{U}_\mathrm{o}\mathbf{h}(t_i) + \mathbf{d}_\mathrm{o}\right) \quad (5d)$$

$$\mathbf{c}_{i+1} \leftarrow \mathbf{f}_{i+1} \odot \mathbf{c}(t_i) + \mathbf{i}_{i+1} \odot \mathbf{z}_{i+1} \quad (6a)$$

$$\bar{\mathbf{c}}_{i+1} \leftarrow \bar{\mathbf{f}}_{i+1} \odot \bar{\mathbf{c}}_i + \bar{\mathbf{i}}_{i+1} \odot \mathbf{z}_{i+1} \quad (6b)$$

$$\boldsymbol{\delta}_{i+1} \leftarrow f\left(\mathbf{W}_\mathrm{d}\mathbf{k}_i + \mathbf{U}_\mathrm{d}\mathbf{h}(t_i) + \mathbf{d}_\mathrm{d}\right) \quad (6c)$$

The vector $\mathbf{k}_i \in \{0,1\}^K$ is the $i^\text{th}$ input: a one-hot encoding of the new event $k_i$, with non-zero value only at the entry indexed by $k_i$. The above formulas will make a discrete update to the LSTM state. They resemble the discrete-time LSTM, but there are two differences. First, the updates do not depend on the "previous" hidden state from just after time $t_{i-1}$, but rather its value $\mathbf{h}(t_i)$ at time $t_i$, after it has decayed for a period of $t_i - t_{i-1}$. Second, equations (6b)–(6c) are new. They define how in future, as $t > t_i$ increases, the elements of $\mathbf{c}(t)$ will continue to deterministically decay (at different rates) from $\mathbf{c}_{i+1}$ toward targets $\bar{\mathbf{c}}_{i+1}$. Specifically, $\mathbf{c}(t)$ is given by (7), which continues to control $\mathbf{h}(t)$ and thus $\lambda_k(t)$ (via (4), except that $i$ has now increased by 1).

$$\mathbf{c}(t) \overset{\text{def}}{=} \bar{\mathbf{c}}_{i+1} + (\mathbf{c}_{i+1} - \bar{\mathbf{c}}_{i+1})\exp\left(-\boldsymbol{\delta}_{i+1}\left(t - t_i\right)\right) \text{ for } t \in (t_i, t_{i+1}] \quad (7)$$

In short, not only does (6a) define the usual cell values $\mathbf{c}_{i+1}$, but equation (7) defines $\mathbf{c}(t)$ on $\mathbb{R}_{>0}$. On the interval $(t_i, t_{i+1}]$, $\mathbf{c}(t)$ follows an exponential curve that begins at $\mathbf{c}_{i+1}$ (in the sense that $\lim_{t \to t_i^+} \mathbf{c}(t) = \mathbf{c}_{i+1}$) and decays toward $\bar{\mathbf{c}}_{i+1}$ (which it would approach as $t \to \infty$, if extrapolated).

A schematic example is shown in Figure 1. As in the previous models, $\lambda_k(t)$ drifts deterministically between events toward some base rate. But the neural version is different in three ways: ① The base rate is not a constant $\mu_k$, but shifts upon each event.[4] ② The drift can be non-monotonic, because the excitatory and inhibitory influences on $\lambda_k(t)$ from different elements of $\mathbf{h}(t)$ may decay at different rates. ③ The sigmoidal transfer function means that the behavior of $\mathbf{h}(t)$ itself is a little more interesting than exponential decay. Suppose that $\mathbf{c}_i$ is very negative but increases toward a target $\bar{\mathbf{c}}_i > 0$. Then $\mathbf{h}(t)$ will stay close to $-1$ for a while and then will rapidly rise past 0. This usefully lets us model a delayed response (e.g. the last green segment in Figure 1).

We point out two behaviors that are naturally captured by our LSTM's "forget" and "input" gates:

- if $\mathbf{f}_{i+1} \approx \mathbf{1}$ and $\mathbf{i}_{i+1} \approx \mathbf{0}$, then $\mathbf{c}_{i+1} \approx \mathbf{c}(t_i)$. So $\mathbf{c}(t)$ and $\mathbf{h}(t)$ will be *continuous* at $t_i$. There is no jump due to event $i$, though the steady-state target may change.
- if $\bar{\mathbf{f}}_{i+1} \approx \mathbf{1}$ and $\bar{\mathbf{i}}_{i+1} \approx \mathbf{0}$, then $\bar{\mathbf{c}}_{i+1} \approx \bar{\mathbf{c}}_i$. So although there may be a jump in activation, it is temporary. The memory cells will decay toward the same steady states as before.

Among other benefits, this lets us fit datasets in which (as is common) some pairs of event types do *not* influence one another. Appendix A.3 explains why all the models in this paper have this ability.

The drift of $\mathbf{c}(t)$ between events controls how the system's expectations about future events change as more time elapses with no event having yet occured. Equation (7) chooses a moderately flexible parametric form for this drift function (see Appendix D for some alternatives). Equation (6a) was designed so that $\mathbf{c}$ in an LSTM could learn to count past events with discrete-time exponential discounting; and (7) can be viewed as extending that to continuous-time exponential discounting.

Our memory cell vector $\mathbf{c}(t)$ is a *deterministic* function of the past history $(\mathcal{H}_i, t - t_i)$.[5] Thus, the event intensities at any time are also deterministic via equation (4). The stochastic part of the model is the random choice—based on these intensities—of *which* event happens next and *when* it happens. The events are in competition: an event with high intensity is likely to happen sooner than an event with low intensity, and whichever one happens first is fed back into the LSTM. If no event type has high intensity, it may take a long time for the next event to occur.

Training the model means learning the LSTM parameters in equations (5) and (6c) along with the other parameters mentioned in this section, namely $s_k \in \mathbb{R}$ and $\mathbf{w}_k \in \mathbb{R}^D$ for $k \in \{1, 2, \ldots, K\}$.

## 4 Algorithms

For the proposed models, the log-likelihood (1) of the parameters turns out to be given by a simple formula—the sum of the log-intensities of the events that happened, at the times they happened, minus an integral of the total intensities over the observation interval $[0, T]$:

$$\ell = \sum_{i:t_i \leq T} \log \lambda_{k_i}(t_i) - \underbrace{\int_{t=0}^{T} \lambda(t)dt}_{\text{call this } \Lambda} \tag{8}$$

The full derivation is given in Appendix B.1. Intuitively, the $-\Lambda$ term (which is $\leq 0$) sums the log-probabilities of infinitely many *non*-events. Why? The probability that there was *not* an event of any type in the infinitesimally wide interval $[t, t + dt)$ is $1 - \lambda(t)dt$, whose log is $-\lambda(t)dt$.

We can locally maximize $\ell$ using any stochastic gradient method. A detailed recipe is given in Appendix B.2, including the Monte Carlo trick we use to handle the integral in equation (8).

If we wish to draw random sequences from the model, we can adopt the thinning algorithm (Lewis and Shedler, 1979; Liniger, 2009) that is commonly used for the Hawkes process. See Appendix B.3.

Given an event stream prefix $(k_1, t_1)$, $(k_2, t_2)$, ..., $(k_{i-1}, t_{i-1})$, we may wish to predict the *time* and *type* of the single next event. The next event's time $t_i$ has density $p_i(t) = P(t_i = t \mid \mathcal{H}_i) = \lambda(t) \exp\left(-\int_{t_{i-1}}^{t} \lambda(s)ds\right)$. To predict a single time whose expected $L_2$ loss is as low as possible, we should choose $\hat{t}_i = \mathbb{E}[t_i \mid \mathcal{H}_i] = \int_{t_{i-1}}^{\infty} tp_i(t)dt$. Given the next event time $t_i$, the most likely type would be $\operatorname{argmax}_k \lambda_k(t_i)/\lambda(t_i)$, but the most likely next event type *without* knowledge of $t_i$ is $\hat{k}_i = \operatorname{argmax}_k \int_{t_{i-1}}^{\infty} \frac{\lambda_k(t)}{\lambda(t)} p_i(t)dt$. The integrals in the preceding equations can be estimated by Monte Carlo sampling much as before (Appendix B.2). For event type prediction, we recommend a paired comparison that uses the same sample of $t$ values for each $k$ in the $\operatorname{argmax}$; this reduces sampling variance and also lets us share the $\lambda(t)$ and $p_i(t)$ computations across all $k$.

## 5 Related Work

The Hawkes process has been widely used to model event streams, including for topic modeling and clustering of text document streams (He et al., 2015; Du et al., 2015a), constructing and inferring network structure (Yang and Zha, 2013; Choi et al., 2015; Etesami et al., 2016), personalized recommendations based on users' temporal behavior (Du et al., 2015b), discovering patterns in social interaction (Guo et al., 2015; Lukasik et al., 2016), learning causality (Xu et al., 2016), and so on.

Recent interest has focused on expanding the expressivity of Hawkes processes. Zhou et al. (2013) describe a self-exciting process that removes the assumption of exponentially decaying influence (as we do). They replace the scaled-exponential summands in equation (2) with learned positive functions of time (the choice of function again depends on $k_i, k$). Lee et al. (2016) generalize the constant excitation parameters $\alpha_{j,k}$ to be stochastic, which increases expressivity. Our model also allows non-constant interactions between event types, but arranges these via deterministic, instead of stochastic, functions of continuous-time LSTM hidden states. Wang et al. (2016) consider non-linear effects of past history on the future, by passing the intensity functions of the Hawkes process through a non-parametric isotonic link function $g$, which is in the same place as our non-linear function $f_k$. In contrast, our $f_k$ has a fixed parametric form (learning only the scale parameter), and is approximately linear when $x$ is large. This is because we model non-linearity (and other complications) with a continuous-time LSTM, and use $f_k$ only to ensure positivity of the intensity functions.

Du et al. (2016) independently combined Hawkes processes with recurrent neural networks (and Xiao et al. (2017a) propose an advanced way of estimating the parameters of that model). However, Du et al.'s architecture is different in several respects. They use standard discrete-time LSTMs without our decay innovation, so they must encode the intervals between past events as explicit numerical inputs to the LSTM. They have only a single intensity function $\lambda(t)$, and it simply decays exponentially toward 0 between events, whereas our more modular model creates separate (potentially transferrable) functions $\lambda_k(t)$, each of which allows complex and non-monotonic dynamics en route to a non-zero steady state intensity. Some structural limitations of their design are that $t_i$ and $k_i$ are conditionally independent given $\mathbf{h}$ (they are determined by separate distributions), and that their model cannot avoid a positive probability of extinction at all times. Finally, since they take

$f = \exp$, the effect of their hidden units on intensity is effectively multiplicative, whereas we take $f = \text{softplus}$ to get an approximately additive effect inspired by the classical Hawkes process. Our rationale is that additivity is useful to capture independent (disjunctive) causes; at the same time, the hidden units that our model adds up can each capture a complex joint (conjunctive) cause.

## 6   Experiments[6]

We fit our various models on several simulated and real-world datasets, and evaluated them in each case by the *log-probability* that they assigned to held-out data. We also compared our approach with that of Du et al. (2016) on their *prediction* task. The datasets that we use in this paper range from one extreme with only $K = 2$ event types but mean sequence length $> 2000$, to the other extreme with $K = 5000$ event types but mean sequence length 3. Dataset details can be found in Table 1 in Appendix C.1. Training details (e.g., hyperparameter selection) can be found in Appendix C.2.

### 6.1   Synthetic Datasets

In a pilot experiment with synthetic data (Appendix C.4), we confirmed that the neural Hawkes process generates data that is not well modeled by training an ordinary Hawkes process, but that ordinary Hawkes data can be successfully modeled by training an neural Hawkes process.

In this experiment, we were not limited to measuring the likelihood of the models on the stochastic event sequences. We also knew the true latent intensities of the generating process, so we were able to directly measure whether the trained models predicted these intensities accurately. The pattern of results was similar.

### 6.2   Real-World Media Datasets

**Retweets Dataset** (Zhao et al., 2015).   On Twitter, *novel* tweets are generated from some distribution, which we do not model here. Each novel tweet serves as the beginning-of-stream event (see Appendix A.2) for a subsequent stream of *retweet* events. We model the dynamics of these streams: how retweets by various types of users ($K = 3$) predict later retweets by various types of users.

Details of the dataset and its preparation are given in Appendix C.5. The dataset is interesting for its temporal pattern. People like to retweet an interesting post soon after it is created and retweeted by others, but may gradually lose interest, so the intervals between retweets become longer over time. In other words, the stream begins in a *self-exciting state*, in which previous retweets increase the intensities of future retweets, but eventually interest dies down and events are less able to excite one another. The decomposable models are essentially incapable of modeling such a phase transition, but our neural model should have the capacity to do so.

We generated learning curves (Figure 2) by training our models on increasingly long prefixes of the training set. As we can see, our self-modulating processes *significantly* outperform the Hawkes process at *all* training sizes. There is no obvious *a priori* reason to expect inhibition or even inertia in this application domain, which explains why the D-SM-MPP makes only a small improvement over the Hawkes process when the latter is well-trained. But D-SM-MPP requires much less data, and also has more stable behavior (smaller error bars) on small datasets. Our neural model is even better. Not only does it do better on the average stream, but its *consistent* superiority over the other two models is shown by the per-stream scatterplots in Figure 3, demonstrating the importance of our model's neural component even with large datasets.

**MemeTrack Dataset** (Leskovec and Krevl, 2014).   This dataset is similar in conception to Retweets, but with many more event types ($K = 5000$). It considers the reuse of fixed phrases, or "memes," in online media. It contains time-stamped instances of meme use in articles and posts from 1.5 million different blogs and news sites. We model how the future occurrence of a meme is affected by its past trajectory across different websites—that is, given one meme's past trajectory across websites, when and where it will be mentioned again.

On this dataset,[7] the advantage of our full neural models was dramatic, yielding cross-entropy per event of around $-8$ relative to the $-15$ of D-SM-MPP—which in turn is *far* above the $-800$ of the

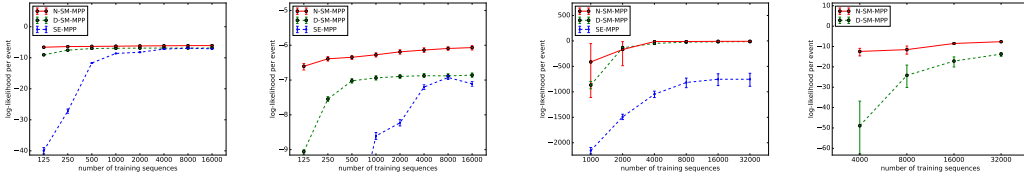

Figure 2: Learning curve (with 95% error bars) of all three models on the Retweets (left two) and MemeTrack (right two) datasets. Our neural model significantly outperforms our decomposable model (right graph of each pair), and both significantly outperform the Hawkes process (left of each pair—same graph zoomed out).

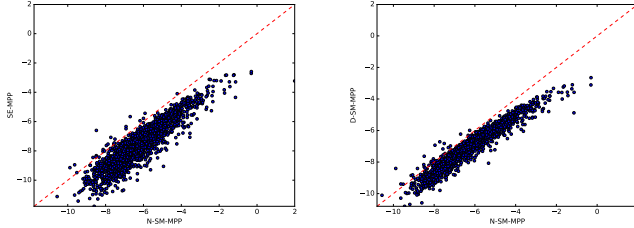

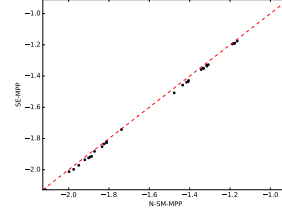

Figure 3: Scatterplots of N-SM-MPP vs. SE-MPP (left) and N-SM-MPP vs. D-SM-MPP (right), comparing the held-out log-likelihood of the two models (when trained on our full Retweets training set) with respect to *each* of the 2000 test sequences. Nearly all points fall to the right of $y = x$, since N-SM-MPP (the neural Hawkes process) is consistently more predictive than our non-neural model and the Hawkes process.

Figure 4: Scatterplot of N-SM-MPP vs. SE-MPP, comparing their log-likelihoods with respect to *each* of the 31 incomplete sequences' test sets. All 31 points fall to the right of $y = x$.

Hawkes process. Figure 2 illustrates the persistent gaps among the models. A scatterplot similar to Figure 3 is given in Figure 13 of Appendix C.6. We attribute the poor performance of the Hawkes process to its failure to capture the latent properties of memes, such as their topic, political stance, or interestingness. This is a form of missing data (section 1), as we now discuss.

As the table in Appendix C.1 indicates, most memes in MemeTrack are uninteresting and give rise to only a short sequence of mentions. Thus the base mention probability is low. An ideal analysis would recognize that if a specific meme has been mentioned several times already, it is *a posteriori* interesting and will probably be mentioned in future as well. The Hawkes process cannot distinguish the interesting memes from the others, except insofar as they appear on more influential websites. By contrast, our D-SM-MPP can partly capture this inferential pattern by using *negative* base rates $\boldsymbol{\mu}$ to create "inertia" (section 3.2.1). Indeed, all 5000 of its learned $\mu_k$ parameters were negative, with values ranging from $-10$ to $-30$, which numerically yields 0 intensity and is hard to excite.

An ideal analysis would also recognize that if a specific meme has appeared mainly on conservative websites, it is *a posteriori* conservative and unlikely to appear on liberal websites in the future. The D-SM-MPP, unlike the Hawkes process, can again partly capture this, by having conservative websites *inhibit* liberal ones. Indeed, 24% of its learned $\alpha$ parameters were negative. (We re-emphasize that this inhibition is merely a predictive effect—probably not a direct causal mechanism.)

And our N-SM-MPP process is even more powerful. The LSTM state aims to learn sufficient statistics for predicting the future, so it can learn hidden dimensions (which fall in $(-1, 1)$) that encode useful posterior beliefs in boolean properties of the meme such as interestingness, conservativeness, timeliness, etc. The LSTM's "long short-term memory" architecture explicitly allows these beliefs to persist indefinitely through time in the absence of new evidence, without having to be refreshed by redundant new events as in the decomposable models. Also, the LSTM's hidden dimensions are computed by sigmoidal activation rather than softplus activation, and so can be used implicitly to perform logistic regression. The flat left side of the sigmoid resembles softplus and can model *inertia* as we saw above: it takes several mentions to establish interestingness. Symmetrically, the flat right side can model *saturation*: once the posterior probability of interestingness is at 80%, it cannot climb much farther no matter how many more mentions are observed.

A final potential advantage of the LSTM is that in this large-$K$ setting, it has fewer parameters than the other models (Appendix C.3), sharing statistical strength across event types (websites) to generalize better. The learning curves in Figure 2 suggest that on small data, the decomposable

(non-neural) models may overfit their $O(K^2)$ interaction parameters $\alpha_{j,k}$. Our neural model only has to learn $O(D^2)$ pairwise interactions among its $D$ hidden nodes (where $D \ll K$), as well as $O(KD)$ interactions between the hidden nodes and the $K$ event types. In this case, $K = 5000$ but $D = 64$. This reduction by using latent hidden nodes is analogous to nonlinear latent factor analysis.

## 6.3 Modeling Streams With Missing Data

We set up an artificial experiment to more directly investigate the missing-data setting of section 1, where we do not observe *all* events during $[0, T]$, but train and test our model just as if we had.

We sampled synthetic event sequences from a standard Hawkes process (just as in our pilot experiment from 6.1), removed all the events of selected types, and then compared the neural Hawkes process (N-SM-MPP) with the Hawkes process (SE-MPP) as models of these censored sequences. Since we took $K = 5$, there were $2^5 - 1 = 31$ ways to construct a dataset of censored sequences. As shown in Figure 4, for *each* of the 31 resulting datasets, training a neural Hawkes model achieves better generalization. Appendix A.4 discusses why this kind of behavior is to be expected.

## 6.4 Prediction Tasks—Medical, Social and Financial

To compare with Du et al. (2016), we evaluate our model on the *prediction* tasks and datasets that they proposed. The Financial Transaction dataset contains long streams of high frequency stock transactions for a single stock, with the two event types "buy" and "sell." The electrical medical records (MIMIC-II) dataset is a collection of de-identified clinical visit records of Intensive Care Unit patients for 7 years. Each patient has a sequence of hospital visit events, and each event records its time stamp and disease diagnosis. The Stack Overflow dataset represents two years of user awards on a question-answering website: each user received a sequence of badges (of 22 different types).

We follow Du et al. (2016) and attempt to predict every held-out event $(k_i, t_i)$ from its history $\mathcal{H}_i$, evaluating the prediction $\hat{k}_i$ with 0-1 loss (yielding an error rate, or ER) and evaluating the prediction $\hat{t}_i$ with L2 loss (yielding a root-mean-squared error, or RMSE). We make minimum Bayes risk predictions as explained in section 4. Figure 8 in Appendix C.7 shows that our model consistently outperforms that of Du et al. (2016) on event type prediction on all the datasets, although for time prediction neither model is consistently better.

## 6.5 Sensitivity to Number of Parameters

Does our method do well because of its flexible nonlinearities or just because it has more parameters? The answer is both. We experimented on the Retweets data with reducing the number of hidden units $D$. Our N-SM-MPP substantially outperformed SE-MPP (the Hawkes process) on held-out data even with very few parameters, although more parameters does even better:

| number of hidden units | Hawkes | 1 | 2 | 4 | 8 | 16 | 32 | 256 |
|---|---|---|---|---|---|---|---|---|
| number of parameters | 21 | 31 | 87 | 283 | 1011 | 3811 | 14787 | 921091 |
| log-likelihood | -7.19 | -6.51 | -6.41 | -6.36 | -6.24 | -6.18 | -6.16 | -6.10 |

We also tried halving $D$ across several datasets, which had negligible effect, always decreasing held-out log-likelihood by $< 0.2\%$ relative.

More information about model sizes is given in Appendix C.3. Note that the neural Hawkes process does not *always* have more parameters. When $K$ is large, we can greatly reduce the number of params below that of a Hawkes process, by choosing $D \ll K$, as for MemeTrack in section 6.2.

## 7 Conclusion

We presented two extensions to the multivariate Hawkes process, a popular generative model of streams of typed, timestamped events. Past events may now either excite *or* inhibit future events. They do so by *sequentially* updating the state of a novel *continuous-time* recurrent neural network (LSTM). Whereas Hawkes sums the time-decaying influences of past events, we instead sum the time-decaying influences of the LSTM nodes. Our extensions to Hawkes aim to address real-world phenomena, missing data, and causal modeling. Empirically, we have shown that both extensions yield a significantly improved ability to predict the course of future events. There are several exciting avenues for further improvements (discussed in Appendix D), including embedding our model within a reinforcement learner to discover causal structure and learn an intervention policy.

## Acknowledgments

We are grateful to Facebook for enabling this work through a gift to the second author. Nan Du kindly helped us by making his code public and answering questions, and the NVIDIA Corporation kindly donated two Titan X Pascal GPUs. We also thank our lab group at Johns Hopkins University's Center for Language and Speech Processing for helpful comments. The first version of this work appeared on arXiv in December 2016.

## Footnotes

[1]More generally, one could allow $0 \le t_1 \le t_2 \le \cdots$, where $t_i$ is a **immediate event** if $t_{i-1} = t_i$ and a **delayed event** if $t_{i-1} < t_i$. It is not too difficult to extend our model to assign positive probability to immediate events, but we will disallow them here for simplicity.

[2]We use one-layer LSTMs with $D$ hidden units in our present experiments, but a natural extension is to use multi-layer ("deep") LSTMs (Graves et al., 2013), in which case $\mathbf{h}(t)$ is the hidden state of the top layer.

[3] The upright-font subscripts i, f, z and o are not variables, but constant labels that distinguish different $\mathbf{W}$, $\mathbf{U}$ and $\mathbf{d}$ tensors. The $\bar{\mathbf{f}}$ and $\bar{\mathbf{i}}$ in equation (6b) are defined analogously to $\mathbf{f}$ and $\mathbf{i}$ but with different weights.

[4] Equations (4b) and (7) imply that after event $i - 1$, the base rate jumps to $f_k(\mathbf{w}^\top(\mathbf{o}_i \odot (2\sigma(2\bar{\mathbf{c}}_i) - 1)))$.

[5] Appendix A.2 explains how our LSTM handles the start and end of the sequence.

[6]Our code and data are available at `https://github.com/HMEIatJHU/neurawkes`.

[7]Data preparation details are given in Appendix C.6.

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
