[Supplementary Material]

# Appendices

*[Supplementary material for Hongyuan Mei & Jason Eisner, "The Neural Hawkes Process: A Neurally Self-Modulating Multivariate Point Process," NIPS 2017.]*

## A  Model Details

In this appendix, we discuss some qualitative properties of our models and give details about how we handle boundary conditions.

### A.1  Discussion of the Transfer Function

As explained in section 3.2, when we allow *inhibition* and *inertia*, we need to pass the total activation through a non-linear **transfer function** $f : \mathbb{R} \to \mathbb{R}_+$ to obtain a positive intensity function. This was our equation (3a), namely $\lambda_k(t) = f(\tilde{\lambda}_k(t))$.

What non-linear function $f$ should we use? The ReLU function $f(x) = \max(x, 0)$ seems at first a natural choice. However, it returns 0 for negative $x$; we need to keep our intensities strictly positive at all times when an event could possibly occur, to avoid infinitely bad log-likelihood at training time or infinite log-loss at test time.

A better choice would be the "softplus" function $f(x) = \log(1 + \exp(x))$, which is strictly positive and approaches ReLU when $x$ is far from 0. Unfortunately, "far from 0" is defined in units of $x$, so this choice would make our model sensitive to the units used to measure time. For example, if we switch the units of $t$ from seconds to milliseconds, then the base intensity $f(\mu_k)$ must become 1000 times lower, forcing $\mu_k$ to be very negative and thus creating a much stronger inertial effect.

To avoid this problem, we introduce a scale parameter $s > 0$ and define $f(x) = s \log(1+\exp(x/s))$. The scale parameter $s$ controls the curvature of $f(x)$, which approaches ReLU as $s \to 0$, as shown in Figure 5. We can regard $f(x)$, $x$, and $s$ as rates, with units of inverse time, so that $f(x)/s$ and $x/s$ are unitless quantities related by softplus. We actually learn a separate scale parameter $s_k$ for each event type $k$, which will adapt to the rate of events of that type.

Figure 5: The softplus function is a soft approximation to a rectified linear unit (ReLU), approaching it as $x$ moves away from 0. We use it to ensure a strictly positive intensity function. We incorporate a scale parameter $s$ that controls the curvature.

### A.2  Boundary Conditions for the LSTM

We initialize the continuous-time LSTM's hidden state to $\mathbf{h}(0) = \mathbf{0}$, and then have it read a special beginning-of-stream (BOS) event $(k_0, t_0)$, where $k_0$ is a special event type (i.e., expanding the

LSTM's input dimensionality by one) and $t_0$ is set to be 0. Then equations (5)–(6) define $\mathbf{c}_1$ (from $\mathbf{c}_0 \overset{\text{def}}{=} \mathbf{0}$), $\bar{\mathbf{c}}_1$, $\boldsymbol{\delta}_1$, and $\mathbf{o}_1$. This is the initial configuration of the system as it waits for the first event to happen: this initial configuration determines the hidden state $\mathbf{h}(t)$ and the intensity functions $\lambda_k(t)$ over $t \in (0, t_1]$

We do not generate the BOS event but only condition on it, which is why the log-likelihood formula (section 2) only sums over $i = 1, 2, \ldots$. This design is well-suited to various settings. In some settings, time 0 is special. For example, if we release children into a carnival and observe the stream of their actions there, then BOS is the release event and no other events can possibly precede it. In other settings, data before time 0 are simply missing, e.g., the observation of a patient starts in midlife; nonetheless, BOS in this case usefully indicates the beginning of the *observed* sequence. In both kinds of settings, the initial configuration just after reading BOS characterizes the model's belief about the unknown state of the true system just after time 0, as it waits for event 1. Computing the initial configuration by explicitly transitioning on BOS ensures that the initial hidden state $\mathbf{h}(0^+) \overset{\text{def}}{=} \lim_{t \to 0^+} \mathbf{h}(t)$ falls in the space of hidden states achievable by LSTM transitions. More important, in future work, we will be able to attach metadata about the sequence as a "mark" to the BOS event (see footnote 13), and the LSTM can learn how these metadata affect the initial configuration.

To allow finite streams, we could optionally choose to identify one of the observable types in $\{1, 2, \ldots, K\}$ as a special end-of-stream (EOS) event after which the stream cannot possibly continue. If the model generates EOS, all intensities are permanently forced to 0—the LSTM is no longer consulted, so it is not necessary for the model parameters to explain why no further events are observed on the interval $[0, T]$: that is, the second term of equation (1) can be omitted. The integral in equation (8) should therefore be taken from $t = 0$ to the time of the EOS event or $T$, whichever is smaller.

## A.3 Closure Under Superposition

Decomposable models have the nice property that they are closed under superposition of event streams. Let $\mathcal{E}$ and $\mathcal{E}'$ be random event streams, on a common time interval $[0, T]$ but over disjoint sets of event types. If each stream is distributed according to a Hawkes process, then their superposition—that is, $\mathcal{E} \cup \mathcal{E}'$ sorted into temporally increasing order—is also distributed according to a Hawkes process. It is easy to exhibit parameters for such a process, using a block-diagonal matrix of $\alpha_{j,k}$ so that the two sets of event types do not influence each other. The closure property also holds for our decomposable self-modulating process, and for the same simple reason.

This is important since in various real settings, some event types tend not to interact. For example, the activities of two people Jay and Kay rarely influence each other,[8] although they are simultaneously monitored and thus form a single observed stream of events. We want our model to handle such situations naturally, rather than insisting that Kay always reacts to what Jay does.

Thus, as section 3.2.2 noted, we have designed our neurally self-modulating process to preserve this ability to insulate event $k$ from event $j$. By setting specific elements of $\mathbf{w}_k$ to 0, one could ensure that the intensity function $\lambda_k(t)$ depends on only a subset $S$ of the LSTM hidden nodes. Then by setting specific LSTM parameters, one would make the nodes in $S$ insensitive to events of type $j$: events of type $j$ should open these nodes' forget gates ($\mathbf{f} = \mathbf{1}$) and close their input gates ($\mathbf{i} = \mathbf{0}$)— as section 3.2.2 suggested—so that their cell memories $\mathbf{c}(t)$ and hidden states $\mathbf{h}(t)$ do not change at all but continue decaying toward their previous steady-state values.[9] Now events of type $j$ cannot affect the intensity $\lambda_k(t)$.

For example, the hidden states in $S$ are affected in the same way when the LSTM reads $(k, 1), (j, 3), (j, 8), (k, 12)$ as when it reads $(k, 1), (k, 12)$, even though the intervals $\Delta t$ between successive events are different. In other words, the architecture "knows" that $2 + 5 + 4 = 11$. The simplicity of this solution is a consequence of how our design does not encode the time intervals

numerically, but only reacts to these intervals indirectly, through the interaction between the timing of events and the spontaneous decay of the hidden states. The memory cells of $S$ decay for a total duration of 11 between the two $k$ events, even if that interval has been divided into subintervals $2 + 5 + 4$.

With this method, we can explicitly construct a superposition process with LSTM state space $\mathbb{R}^{d+d'}$—the cross product of the state spaces $\mathbb{R}^d$ and $\mathbb{R}^{d'}$ of the original processes—in which Kay's events are not influenced at all by Jay's.

If we know *a priori* that particular event types interact only weakly, we can impose an appropriate prior on the neural Hawkes parameters. And in future work with large $K$, we plan to investigate the use of sparsity-inducing regularizers during parameter estimation, to create an inductive bias toward models that have limited interactions, without specifying which particular interactions are present.

Superposition is a formally natural operation on event streams. It barely arises for ordinary sequence models, such as language models, since the superposition of two sentences is not well-defined unless all of the words carry distinct real-valued timestamps. However, there is an analogue from formal language theory. The "shuffle" of two sentences is defined to be the set of *possible* interleavings of their words—i.e., the set of superpositions that could result from assigning increasing timestamps to the words of each sentence, without duplicates. It is a standard exercise to show that regular languages are closed under shuffle. This is akin to our remark that neural-Hawkes-distributed random variables are closed under superposition, and indeed uses a similar cross-product construction on the finite-state automata. An important difference is that the shuffle construction does not require disjoint alphabets in the way that ours requires disjoint sets of event types. This is because finite-state automata allow nondeterministic state transitions and our processes do not.

## A.4 Missing Data Discussion

We discussed the case of missing data in section 1. Suppose the true complete-data distribution $p^*$ is itself an unknown neural Hawkes process. As section 1 pointed out, a sufficient statistic for prediction from the incompletely observed past would be the posterior distribution over the true hidden neural state $\mathbf{t}$ of the unknown process, which was reached by reading the *complete* past. We would ideally obtain our predictions by correctly modeling the missing observations and integrating over them. However, inference would be computationally quite expensive even if $p^*$ were known, to say nothing of the case where $p^*$ is unknown and we must integrate over its parameters as well.

We instead train a neural model that attempts to bypass these problems. The hope is that our model's hidden state, after it reads only the observed *incomplete* past, will be nearly as predictive as the posterior distribution above.

We can illustrate the goal with reference to the experiment in section 6.3. There, the true complete-data distribution $p^*$ happened to be a classical Hawkes process, but we censored some event types. We then modeled the observed incomplete sequence as if it were a complete sequence. In this setting, a Hawkes process will in general be unable to fit the data well, which is why the neural Hawkes process has an advantage in all 31 experiments.

What goes wrong with using the Hawkes model? Suppose that in the true Hawkes model $p^*$, type 1 is rare but strongly excites type 2 and type 3, which do not excite themselves or each other. Type 1 events are missing in the observed sequence.

What is the correct predictive distribution in this situation (with knowledge of $p^*$)? Seeing lots of type 2 events in a row suggests that they were preceded by a (single) missing type 1 event, which predicts a higher intensity for type 3 in future. The more type 2 events we see, the surer we are that there was a type 1 event, but we doubt that there were multiple type 1 events, so the predicted intensity of type 3 is expected to increase sublinearly as $P(\text{type} = 1)$ approaches 1.

As neural networks are universal function approximators , a neural Hawkes model may be able to recognize and fit this sublinear behavior in the incomplete training data. However, if we fit only a Hawkes model to the incomplete training data, it would have to posit that type 2 excites type 3 directly, so the predicted intensity of type 3 would incorrectly increase linearly with the number of type 2 events.

# B   Algorithmic Details

In this appendix, we elaborate on the details of algorithms.

## B.1   Likelihood Function

For the proposed models, given complete observations of an event stream over the time interval $[0, T]$, the log-likelihood of the parameters turns out to be given by the simple formula shown in section 4. We start by giving the full derivation of that formula, repeated here:

$$\ell = \sum_{i:t_i \leq T} \log \lambda_{k_i}(t_i) - \underbrace{\int_{t=0}^{T} \lambda(t)dt}_{\text{call this } \Lambda} \tag{8}$$

First, we define $N(t) = |\{h : t_h \leq t\}|$ to be the count of events (of any type) preceding time $t$. So given the past history $\mathcal{H}_i$, the number of events in $(t_{i-1}, t]$ is denoted as $\Delta N(t_{i-1}, t) \overset{\text{def}}{=} N(t) - N(t_{i-1})$. Let $T_i > t_{i-1}$ be the random variable of the next event time and let $K_{i+1}$ be the random variable of the next event type. The cumulative distribution function and probability density function of $T_i$ (conditioned on $\mathcal{H}_i$) are given by:

$$F(t) = P(T_i \leq t) = 1 - P(T_i > t) \tag{9a}$$
$$= 1 - P(\Delta N(t_{i-1}, t) = 0) \tag{9b}$$
$$= 1 - \exp\left(-\int_{t_{i-1}}^{t} \lambda(s)ds\right) \tag{9c}$$
$$= 1 - \exp\left(\Lambda(t_{i-1}) - \Lambda(t)\right) \tag{9d}$$
$$f(t) = \exp\left(\Lambda(t_{i-1}) - \Lambda(t)\right)\lambda(t) \tag{9e}$$

where $\Lambda(t) = \int_0^t \lambda(s)ds$ and $\lambda(t) = \sum_{k=1}^{K} \lambda_k(t)$.

Moreover, given the past history $\mathcal{H}_i$ and the next event time $t_i$, the distribution of $k_i$ is given by:

$$P(K_i = k_i \mid t_i) = \frac{\lambda_{k_i}(t_i)}{\lambda(t_i)} \tag{10}$$

Therefore, we can derive the likelihood function as follows:

$$\mathcal{L} = \prod_{i:t_i \leq T} \mathcal{L}_i = \prod_{t_i \leq T} \{f(t_i)P(K_i = k_i \mid t_i)\} \tag{11a}$$
$$= \prod_{i:t_i \leq T} \{\exp\left(\Lambda(t_{i-1}) - \Lambda(t_i)\right)\lambda_{k_i}(t_i)\} \tag{11b}$$

and

$$\ell \overset{\text{def}}{=} \log \mathcal{L} \tag{12a}$$
$$= \sum_{i:t_i \leq T} \log \lambda_{k_i}(t_i) - \sum_{i:t_i \leq T} (\Lambda(t_i) - \Lambda(t_{i-1})) \tag{12b}$$
$$= \sum_{i:t_i \leq T} \log \lambda_{k_i}(t_i) - \Lambda(T) \tag{12c}$$
$$= \sum_{i:t_i \leq T} \log \lambda_{k_i}(t_i) - \int_{t=0}^{T} \lambda(t)dt \tag{12d}$$

## B.2   Monte Carlo Gradient and Training Speed

We can locally maximize the log-likelihood $\ell$ from equation (8) using any stochastic gradient method. For this, we need to be able to get an unbiased estimate of the gradient $\nabla \ell$ with respect to the model parameters. This is straightforward to obtain by back-propagation. The trick

**Algorithm 1** Integral Estimation (Monte Carlo)

---

**Input:** interval $[0, T]$; model parameters and
 events $(k_1, t_1), \dots$ for determining $\lambda_j(t)$
$\Lambda \leftarrow 0; \nabla\Lambda \leftarrow \mathbf{0}$
**for** $N$ samples : $\quad\quad\quad\quad\quad\quad\quad\quad\quad\quad\quad\quad\quad$ ▷ *e.g., take $N > 0$ proportional to $T$*
 draw $t \sim \mathrm{Unif}(0, T)$
 **for** $j \leftarrow 1$ **to** $K$ :
  $\Lambda \mathrel{+}= \lambda_j(t) \quad\quad\quad\quad\quad\quad\quad\quad\quad\quad$ ▷ *via current model parameters*
  $\nabla\Lambda \mathrel{+}= \nabla\lambda_j(t) \quad\quad\quad\quad\quad\quad\quad\quad\quad$ ▷ *via back-propagation*
$\Lambda \leftarrow T\Lambda/N; \nabla\Lambda \leftarrow T\nabla\Lambda/N \quad\quad\quad\quad\quad$ ▷ *weight the samples*
**return** $(\Lambda, \nabla\Lambda)$

---

for handling the integral in equation (8) is that the single function evaluation $T\lambda(t)$ at a random $t \sim \mathrm{Unif}(0, T)$ gives an unbiased estimate of the entire integral—that is, its expected value is $\Lambda$. Its gradient via back-propagation is therefore a unbiased estimate of $\nabla\Lambda$ (since gradient commutes with expectation). The Monte Carlo algorithm in Algorithm 1 averages over several samples to reduce the variance of this noisy estimator.

Each step of Adam training computes the gradient on a training sequence. With $P$ params, this takes time $O(IP)$ for Hawkes and $O((I+M)P)$ for neural Hawkes, if $I$ is the number of observed events and $M$ is the number of samples used to estimate the integral. We take $M = O(I)$ in practice (see Appendix C.2), so we have runtime $O(IP)$ like Hawkes.

Note that our stochastic gradient is unbiased for any $M$; large $M$ merely reduces its variance. The gradient for the Hawkes process has 0 variance, since it has analytical form and does not require sampling at all.

## B.3 Thinning Algorithm for Sampling Sequences

If we wish to draw sequences from the self-modulating models of 3.2, we can adopt the thinning algorithm (Lewis and Shedler, 1979; Liniger, 2009) that is commonly used for the multivariate Hawkes process, as shown in Algorithm 2. We explain the algorithm here and illustrate its conception in Figure 6.

Suppose we have already sampled the first $i-1$ events. The $K$ event types are now in a race to see who generates the next event. (Typically, the winning type will have relatively high intensity.) In our model, that next event will join the multivariate event stream as $(k_i, t_i)$, whereupon it updates the LSTM state and thus modulates the subsequent intensities that will be used to sample event $i + 1$.

How do we conduct the race? For each event type $k$, let the function $\lambda_k^i : (t_{i-1}, \infty) \to \mathbb{R}_{\geq 0}$ map each time $t$ to the intensity $\lambda_k^i(t)$ that our model will define at time $t$ provided that event $i$ has not yet happened in the interval $(t_{i-1}, t)$. For each $k$ independently, we draw the time $t_{i,k}$ of the next event from the non-homogeneous Poisson process over $(t_{i-1}, \infty)$ whose intensity function is $\lambda_k^i$. We then take $t_i = \min_k t_{i,k}$ and $k_i = \mathrm{argmin}_k t_{i,k}$. That is, we keep just the earliest of the $K$ events. We cannot keep the rest because they are not correctly distributed according to the new intensities as updated by the earliest event.

But how do we draw the next event time $t_{i,k}$ from the non-homogeneous Poisson process given by $\lambda_k^i$? Recall from 3.1 that a draw from such a point process is actually a whole *set* of times in $(t_{i-1}, \infty)$: we will take $t_{i,k}$ to be the earliest of these. In theory, this set is drawn by *independently* choosing at each time $t \in (t_{i-1}, \infty)$, with infinitesimal probability proportional to $\lambda_k^i(t)$, whether an event occurs. One could do this by *independently* applying rejection sampling at each time $t$: choose with larger probability $\lambda^*$ whether a "proposed event" occurs at time $t$, and if it does, accept the proposed event with probability only $\lambda_k^i(t)/\lambda^* \leq 1$. This is equivalent to simultanously drawing a set of proposed times from a *homogenous* Poisson process with constant rate $\lambda^*$, and then "thinning" that proposed set, as illustrated in Figure 6. This approach helps because it is easy to draw from the homogenous process: the intervals between successive proposed events are IID $\mathrm{Exp}(\lambda^*)$, so it is easy to sample the events in sequence. The inner **repeat** loop in Algorithm 2 lazily carries out just enough of this infinite homogenous draw from $\lambda^*$ to determine the time $t_{i,k}$ of the earliest *accepted* event, which is the earliest event in the non-homogeneous draw from $\lambda_k^i$, as desired.

Figure 6: Sampling the next event, using the same visual notation as in Figure 1. The $x$ axis shows a prefix of the infinite interval $(t_{i-1}, \infty)$. In the first graph, *gold* events are proposed from a homogeneous Poisson process with intensity $\lambda^*$ (gold straight line). In the second graph, the purple curve $\lambda_1^i$ randomly accepts some of these gold events, with probability $\lambda_1^i(t)/\lambda^*$ for the event at time $t$; here it accepts three of the ones shown and rejects the others. In the third graph, the surviving type-1 events (purple squares) are interleaved with the surviving type-2 events (green pentagons). The next event is the earliest one among these surviving candidates. In practice, these sequences are constructed lazily so that we find only the earliest surviving event of each type. This is possible because the inter-arrival times between gold proposed events are distributed as $\mathrm{Exp}(\lambda^*)$, making it straightforward to enumerate any finite prefix of a random infinite gold sequence.

---

**Algorithm 2** Data Simulation (thinning algorithm)

---

  **Input:** interval $[0,T]$; model parameters
  $t_0 \leftarrow 0; i \leftarrow 1$
  **while** $t_{i-1} < T$ :                                      ▷ *draw event i, as it might fall in [0,T]*
    **for** $k = 1$ **to** $K$ :                                ▷ *draw "next" event of each type*
      find upper bound $\lambda^* \geq \lambda_k^i(t)$ for all $t \in (t_{i-1}, \infty)$
      $t \leftarrow t_{i-1}$
      **repeat**
        draw $\Delta \sim \mathrm{Exp}(\lambda^*), u \sim \mathrm{Unif}(0,1)$
        $t\mathrel{+}=\Delta$                                  ▷ *time of next proposed event*
      **until** $u\lambda^* \leq \lambda_k^i(t)$                    ▷ *accept proposal with prob* $\frac{\lambda_k^i(t)}{\lambda^*}$
      $t_{i,k} \leftarrow t$
    $t_i \leftarrow \min_k t_{i,k}; k_i \leftarrow \operatorname{argmin}_k t_{i,k}$             ▷ *earliest event wins*
    $i \leftarrow i + 1$
  **return** $(k_1, t_1), \ldots (k_{i-1}, t_{i-1})$

---

Finally, how do we construct the upper bound $\lambda^*$ on $\lambda_k^i$? Recall that both of our self-modulating models (equations (3a) and (4a)) define $\lambda_k^i = f_k(\tilde{\lambda}_k^i)$, where $f_k$ is monotonically non-decreasing. In both cases, $\tilde{\lambda}_k^i$ is a sum of *bounded* functions on $(t_{i-1}, \infty)$ (equations (3b) and (4)). In other words, we can express $\tilde{\lambda}_k^i(t)$ as $\mu + g_1(t) + \cdots + g_n(t)$. We can therefore replace each $g$ function by its upper bound to obtain $\lambda^* = f_k(\mu + \max_t g_1(t) + \cdots + \max_t g_n(t))$, in which the argument to $f_k$ is a finite constant.

Specifically, in equation (3b), each summand $\alpha_{k_h,k} \exp(-\delta_{k_h,k}(t - t_i))$ is upper-bounded by $\max(\alpha_{k_h,k}, 0)$. In equation (4), each summand $w_{kd} h_d(t) = w_{kd} \cdot o_{id} \cdot (2\sigma(2c_d(t)) - 1)$ is upper-bounded by $\max_{c \in \{c_{id}, \bar{c}_{id}\}} w_{kd} \cdot o_{id} \cdot (2\sigma(2c) - 1)$. Note that the coefficients $\alpha_{k_i,k}$ and $w_{kd}$ may be either positive or negative.

While Algorithm 2 is classical and intuitive, we also implemented a more efficient variant. Instead of drawing the next event from each of $K$ *different* non-homogeneous Poisson processes and keeping the earliest, we can construct a *single* non-homogenous Poisson process with aggregate intensity function $\lambda^i(t) = \sum_{k=1}^K \lambda_k^i(t)$ over $(t_{i-1}, \infty)$. An upper bound $\lambda^*$ on this aggregate function can be obtained by summing the upper bounds on the individual $\lambda_k^i$ functions. We then use the thinning algorithm only to sample the next event time $t_i$ from this aggregate process $\lambda^i$. Finally, we "dis-aggregate" by choosing $k_i$ from the distribution $p(k \mid t_i) = \lambda_k^i(t_i)/\lambda^i(t_i)$.[10] This is equivalent to Algorithm 2. In terms of Figure 6, this more efficient version enumerates a gold sequence that is the union of the $K$ gold sequences, and stops with the first accepted gold event. Thus, whereas Figure 6

| DATASET | $K$ | # OF EVENT TOKENS | | | SEQUENCE LENGTH | | |
|---------|-----|-------|-----|------|-----|------|-----|
| | | TRAIN | DEV | TEST | MIN | MEAN | MAX |
| SYNTHETIC | 5 | $\approx 480449$ | $\approx 60217$ | $\approx 60139$ | 20 | $\approx 60$ | 100 |
| RETWEETS | 3 | 1739547 | 215521 | 218465 | 50 | 109 | 264 |
| MEMETRACK | 5000 | 93267 | 14932 | 15440 | 1 | 3 | 31 |
| MIMIC-II | 75 | $\approx 1946$ | $\approx 228$ | $\approx 245$ | 2 | 4 | 33 |
| STACKOVERFLOW | 22 | $\approx 343998$ | $\approx 39247$ | $\approx 97168$ | 41 | 72 | 736 |
| FINANCIAL | 2 | $\approx 298710$ | $\approx 33190$ | $\approx 82900$ | 829 | 2074 | 3319 |

Table 1: Statistics of each dataset. We write "$\approx N$" to indicate that $N$ is the average value over multiple splits of one dataset (MIMIC-II, Stack Overflow, Financial Transaction); the variance is small in each such case.

| DATASET | $K$ | $D$ | # OF MODEL PARAMETERS | | |
|---------|-----|-----|-------|---------|---------|
| | | | SE-MPP | D-SM-MPP | N-SM-MPP |
| SYNTHETIC | 5 | 256 | 55 | 60 | 922117 |
| RETWEETS | 3 | 256 | 21 | 24 | 921091 |
| MEMETRACK | 5000 | 64 | 50005000 | 50010000 | 702856 |

Table 2: Size of each trained model on each dataset. The number of parameters of neural Hawkes process is followed by the number of hidden nodes $D$ in its LSTM (chosen automatically on dev data).

had to propose two type-1 events in order to get the first accepted type-1 event (the leftmost purple event), the more efficient version would not have had to spend time proposing either of those, because an earlier proposed event (the leftmost green event) had already been accepted and determined to be of type 2.

# C   Experimental Details

In this appendix, we elaborate on the details of data generation, processing, and experimental results.

## C.1   Dataset Statistics

Table 1 shows statistics about each dataset that we use in this paper.

## C.2   Training Details

We used a single-layer LSTM (Graves, 2012) in section 3.2.2, selecting the number of hidden nodes from a small set $\{64, 128, 256, 512, 1024\}$ based on the performance on the dev set of each dataset. We empirically found that the model performance is robust to these hyperparameters.

When estimating integrals with Monte Carlo sampling, $N$ is the number of sampled negative observations in Algorithm 1, while $I$ is the number of positive observations. In practice, setting $N = I$ was large enough for stable behavior, and we used this setting during training. For evaluation on dev and test data, we took $N = 10\,I$ for extra accuracy, or $N = I$ when $I$ was very large.

For learning, we used the Adam algorithm with its default settings (Kingma and Ba, 2015). Adam is a stochastic gradient optimization algorithm that continually adjusts the learning rate in each dimension based on adaptive estimates of low-order moments. Our training objective was unregularized log-likelihood.[11] We initialized the Hawkes process parameters and $s_k$ scale factors to 1, and all other non-LSTM parameters (section 3.2.2) to small random values from $\mathcal{N}(0, 0.01)$. We performed early stopping based on log-likelihood on the held-out dev set.

Figure 7: Log-likelihood (reported in nats per event) of each model on held-out synthetic data. Rows (top-down) are log-likelihood on the entire sequence, time interval, and event type. On each row, the figures (from left to right) are datasets generated by SE-MPP, D-SM-MPP and N-SM-MPP. In each figure, the models (from left to right) are Oracle, SE-MPP, D-SM-MPP and N-SM-MPP. Larger values are better. Note that log-likelihood for continuous variables can be positive, since it uses the log of a probability density that may be $> 1$.

## C.3 Model Sizes

The size of each trained model on each dataset is shown in Table 2. Our neural model has many parameters for expressivity, but it actually has considerably fewer parameters than the other models in the large-$K$ setting (MemeTrack).

## C.4 Pilot Experiments on Simulated Data

Our hope is that the neural Hawkes process is a flexible tool that can be used to fit naturally occurring data. As mentioned in section 6.1, we first checked that we could successfully fit data generated from *known* distributions. That is, when the generating distribution actually fell within our model family, could our training procedure recover the distribution in practice? When the data came from a decomposable process, could we nonetheless train our neural process to fit the distribution well?

We used the thinning algorithm (Appendix B.3) to sample event streams from different processes with randomly generated parameters: (a) a standard Hawkes process (SE-MPP, section 3.1), (b) our decomposable self-modulating process (D-SM-MPP, section 3.2.1), (c) our neural self-modulating processes (N-SM-MPP, section 3.2.2). We then tried to fit each dataset with all these models.[12]

The results are shown in Figure 7. We found that all models were able to fit the (a) and (b) datasets well with no statistically significant difference among them, but that the (c) models were substantially and significantly better at fitting the (c) datasets. In all cases, the (c) models were able to obtain a low KL divergence from the true generating model (the difference from the oracle column). This result suggests that the neural Hawkes process may be a wise choice: it introduces extra expressive power that is sometimes necessary and does not appear (at least in these experiments) to be harmful when it is not necessary.

We used Algorithm 2 to sample event streams from three different processes with randomly generated parameters: (a) a standard Hawkes process (SE-MPP), (b) our decomposable self-modulating process (D-SM-MPP), (c) our neural self-modulating processes (N-SM-MPP). We then tried to fit each dataset with all these models.

For each dataset, we took $K = 5$ as the number of event types. To generate each event sequence, we first chose the sequence length $I$ (number of event tokens) uniformly from $\{20, 21, 22, \ldots, 100\}$ and then used the thinning algorithm to sample the first $I$ events over the interval $[0, \infty)$. For subsequent training or testing, we treated this sequence (appropriately) as the complete set of events observed on the interval $[0, T]$ where $T = t_I$, the time of the last generated event. For each dataset, we generate 8000, 1000 and 1000 sequences for the training, dev, and test sets respectively.

For SE-MPP, we sampled the parameters as $\mu_k \sim \text{Unif}[0.0, 1.0]$, $\alpha_{j,k} \sim \text{Unif}[0.0, 1.0]$, and $\delta_{j,k} \sim \text{Unif}[10.0, 20.0]$. The large decay rates $\delta_{j,k}$ were needed to prevent the intensities from blowing up as the sequence accumulated more events. For D-SM-MPP, we sampled the parameters as $\mu_k \sim \text{Unif}[-1.0, 1.0]$, $\alpha_{j,k} \sim \text{Unif}[-1.0, 1.0]$, and $\delta_{j,k} \sim \text{Unif}[10.0, 20.0]$. For N-SM-MPP, we sampled parameters from $\text{Unif}[-1.0, 1.0]$.

The results are shown in Figure 7, including log-likelihood (reported in nats per event) on the sequences and the breakdown of time interval and event types.

Another interesting question is whether the trained neural Hawkes model accurately predicts the real-valued *intensities*, since for the synthetic data we actually know the intensities. This is a more direct evaluation of whether the model is accurately recovering the dynamics of the underlying generative process. Here we compared only SE-MPP and N-SM-MPP.

All types behaved similarly, so we report only averages over the $K$ types. For both processes (a) and (c), the true intensity's variance was about 30% of the squared mean intensity. Thus, the intensity changes enough over time that predicting it at particular times is not a trivial challenge. To determine how well a model predicted the true intensity function, we measured the mean squared error (MSE) of predicted intensity at a large sample of times in the held-out test seqs, and report the MSE here as a percentage of the *variance* of the true intensity. By this construction, a simple baseline of predicting each event type's mean intensity at all times would get 100% MSE.

Both the Hawkes and neural-Hawkes models predict the Hawkes intensities (a) accurately, at 1% MSE. This is similar to the leftmost column of Figure 7, where both models essentially achieved oracle performance. By contrast, for the complex neural Hawkes intensities (c), the neural Hawkes model achieves 9% MSE (still quite good) whereas Hawkes does far worse at 70% MSE. This is similar to the rightmost column of Figure 7, where the neural Hawkes model approached oracle performance but the Hawkes model did much worse.

## C.5 Retweet Dataset Details

The Retweets dataset (section 6.2) includes 166076 retweet sequences, each corresponding to some original tweet. Each retweet event is labeled with the retweet time relative to the original tweet creation, so that the time of the original tweet is 0. (The original tweet serves as the beginning-of-stream (BOS) marker as explained in Appendix A.2.) Each retweet event is also marked with the number of followers of the retweeter. As usual, we assume that these 166076 streams are drawn independently from the same process, so that retweets in different streams do not affect one another.

Unfortunately, the dataset does not specify the identity of each retweeter, only his or her popularity. To distinguish different kinds of events that might have different rates and different influences on the future, we divide the events into $K = 3$ types: retweets by "small," "medium" and "large" users. Small users have fewer than 120 followers (50% of events), medium users have fewer than 1363 (45% of events), and the rest are large users (5% events). Given the past retweet history, our model must learn to predict how soon it will be retweeted again and how popular the retweeter is (i.e., which of the three categories).

We randomly sampled disjoint train, dev and test sets with 16000, 2000 and 2000 sequences respectively. We truncated sequences to a maximum length of 264, which affected 20% of them. For computing training and test likelihoods, we treated each sequence as the complete set of events observed on the interval $[0, T]$, where 0 denotes the time of the original tweet (which is not included in the sequence) and $T$ denotes the time of the last tweet in the (truncated) sequence.

Figure 8: Prediction results on Financial Transactions, MIMIC-II, and Stack Overflow datasets (from left to right). Error bars show standard deviation over 5 experiments with different train-dev-test splits. For prediction of the types $k_i$ (top row), our method achieved lower error in 4/5, 5/5, and 5/5 of the experiments. For prediction of the times $t_i$ (bottom row), our method achieved lower error in 5/5, 2/5, and 0/5 of the experiments.

Figure 9 shows the learning curves of all the models, broken down by the log-probabilities of the event types and the time intervals separately. The scatterplot Figure 10 is a copy of Figure 3, and Figure 11 breaks down the log-likelihood by event type and time interval.

## C.6 MemeTrack Dataset Details

The MemeTrack dataset (section 6.2) contains time-stamped instances of meme use in articles and posts from 1.5 million different blogs and news sites, spanning 10 months from August 2008 till May 2009, with several hundred million documents.

As in Retweets, we decline to model the appearance of novel memes. Each novel meme serves as the BOS event for a stream of mentions on other websites, which we do model. The $K$ event types correspond to the different websites. Given one meme's past trajectory across websites, our model must learn to predict how soon it will be mentioned again and where.

We used the version of the dataset processed by Gomez Rodriguez et al. (2013), which selected the top 5000 websites in terms of the number of memes they mentioned. We truncated sequences to a maximum length of 32, which affected only 1% of them. We randomly sampled disjoint train, dev and test sets with 32000, 5000 and 5000 sequences respectively, treating them as before.

Because our current implementation does not allow for a marked BOS event (see Appendix A.2), we currently ignore where the novel meme was originally posted, making the unfortunate assumption that the stream of websites is independent of the originating website. Even worse, we must assume that the stream of websites is independent of the actual text of the meme. However, as we see, our novel models have some ability to recover from these forms of missing data.

Figure 12 shows the learning curves of the breakdown of log-likelihood with the same format as Figure 9. Figures 13 and 14 show the scatterplots in the same format as Figures 10 and 11.

## C.7 Prediction Task Details

Finally, we give further details of the prediction experiments from section 6.4. To avoid tuning on the test data, we split the original training set into a new training set and a held-out dev set. We train our neural model and that of Du et al. (2016) on the new training set, and choose hyper-parameters on the held-out dev set. Following Du et al. (2016), we consider three datasets, and use five different train-dev-test splits of each dataset to generate the experimental results in Figure 8. (None of the test sets' examples were used during manual development of our system.)

Figure 9: Learning curves (with $95\%$ error bars) of all these models on the Retweets dataset, broken down by the log-probabilities of just the event types (left graph) and just the time intervals (right graph).

Figure 10: A larger copy of Figure 3, repeated here for convenience.

Figure 11: Scatterplots of N-SM-MPP vs. SE-MPP on Retweets. Same comparison as the left graph in Figure 10, but broken down by the log-probabilities of the event types (left graph) and the time intervals (right graph).

Figure 12: Learning curve (with 95% error bars) of all three models on the MemeTrack dataset, broken down by the log-probabilities of the event types (left graph) and the time intervals (right graph).

Figure 13: Scatterplot of N-SM-MPP vs. SE-MPP (left graph) and vs. D-SM-MPP (right graph) on MemeTrack. N-SM-MPP outperforms D-SM-MPP on 93.02% of the test sequences. This is not obvious from the plot, because almost all of the 5000 points are crowded near the upper right corner. Most of the visible points are outliers where N-SM-MPP performs unusually badly—and D-SM-MPP typically does even worse.

Figure 14: Scatterplots of N-SM-MPP vs. SE-MPP on MemeTrack. Same comparison as the left graph of Figure 13, but broken down by the log-probabilities of the event types (left graph) and the time intervals (right graph).

# D  Ongoing and Future Work

We are currently exploring several extensions to deal with more complex datasets. Based on our survey of existing datasets, we are particularly interested in handling:

- immediate events ($t_{i-1} = t_i$), as discussed in footnote 1
- "baskets" of events (several events that are recorded as occuring simultaneously but without a specified order, e.g., the purchase of an entire shopping cart)
- hard constraints on the event type sequence $k_1, k_2, \ldots$
- marked events[13] and annotated events[14]
- causation by external events (artificial clock ticks, periodic holidays, weather)
- richer drift functions[15]
- hybrid of D-SM-MPP and N-SM-MPP, allowing direct influence from past events
- multiple agents each with their own state, who observe one another's actions (events)

More important, we are interested in modeling causality. The current model might pick up that a hospital visit elevates the instantaneous probability of death, but this does not imply that a hospital visit *causes* death. (In fact, the severity of an earlier illness is usually the cause of both.)

A model that can predict the result of interventions is called a causal model. Our model family can naturally be used here: any choice of parameters defines a generative story that follows the arrow of time, which can be interpreted as a causal model in which patterns of earlier events *cause* later events to be more likely. Such a causal model predicts how the distribution over futures would change if we intervened in the stream of events.

In general, one cannot determine the parameters of a causal model based on purely observational data (Pearl, 2009). Thus, in future, we plan to determine such parameters through randomized experiments by deploying our model family as an environment model within reinforcement learning. A reinforcement learning agent *tests* the effect of random interventions to discover their effect (exploration) and thus orchestrate more rewarding futures (exploitation).

In our setting, the agent is able to stochastically insert or suppress certain event types and observe the effect on subsequent events. Then our LSTM-based model will discover the causal effects of such actions, and the reinforcement learner will discover what actions it can take to affect future reward. Ultimately this could be a vehicle for personalized medical decision-making. Beyond the medical domain, a quantified-self smartphone app may intervene by displaying fine-grained advice on eating, sleeping, exercise, and travel; a charitable agency may intervene by sending a social worker to provide timely counseling or material support; a social media website may increase positive engagement by intelligently distributing posts; or a marketer may stimulate consumption by sending more targeted advertisements.

## Footnotes

[8] Their surnames might be Box and Cox, after the 19th-century farce about a day worker and a night worker unknowingly renting the same room. But any pair of strangers would do.

[9] To be precise, we can achieve this arbitrarily closely, but not exactly, because a standard LSTM gate cannot be fully opened or closed. The openness is traditionally given by a sigmoid function and so falls in $(0, 1)$, never achieving 1 or 0 exactly unless we are willing to set parameters to $\pm\infty$. In practice this should not be an issue because relatively small weights can drive the sigmoid function extremely close to 1 and 0—in fact, $\sigma(37) = 1$ in 64-bit floating-point arithmetic.

[10]In practice, acceptance and disaggregation can be combined into a single step. That is, each successive event $t$ proposed from the homogeneous Poisson$(\lambda^*)$ process is either kept as type $k$, with probability $\lambda_k^i(t)/\lambda^*$, or rejected, with probability $1 - \lambda^i(t)/\lambda^*$. If it is accepted, we have found our next event $(k_i, t_i)$. If it is rejected, we increment $t$ by $\Delta \sim \mathrm{Exp}(\lambda^*)$ to get the next proposed event.

[11]$L_2$ regularization did not appear helpful in pilot experiments, at least for our dataset size and when sharing a single regularization coefficient among all parameters.

[12]Details of data generation can be found in Appendix C.4.

[13]A "mark" is some structured data attached to an event: for example, the textual content associated with a tweet, or the medical records associated with a doctor visit. The model should predict the marks from each event and its underlying hidden state, and they should be fed back into the LSTM as additional input.

[14]Humans may be asked to classify the events in an event stream or the relationships among its events. Unlike marks, these annotations are not involved in the process that generates the event stream, and so are not fed into the LSTM as input. Rather, they are assumed to be generated *post hoc* by the human from the entire observed stream—and may depend on the human's implicit reconstruction of the hidden states. We can use any available annotations to help reconstruct the hidden states (Zaidan and Eisner, 2008), if we model them as stochastic functions of the hidden states. In particular, annotations on the training data serve as side information to improve training of the model. As a simple example, an annotation of the training event $(k_i, t_i)$ could be assumed to depend also on the subsequent LSTM state $\mathbf{h}(t_i^+) \stackrel{\text{def}}{=} \lim_{t \to t_i^+} \mathbf{h}(t)$.

[15]We expect the exponential drift in equation (7) to be expressive enough in most settings. In principle, however, one might want to allow periodic fluctuation of the intensity between events, say by using a *complex* exponential in (7). Another way to increase expressivity would be to compute drift using the LSTM itself, by injecting special "clock tick" events into the input stream at regular intervals (compare Xiao et al., 2017b). Each clock tick event $(k_i, t_i)$ causes a rich nonlinear update of the LSTM state via equations (5)–(6), except that it should always set $\mathbf{c}_{i+1} = \mathbf{c}(t_i)$ for continuity. In this design, the interval between ordinary events is modeled piecewise—it is divided up into short pieces by the clock ticks, with $\mathbf{c}(t)$ on each piece modeled using our current function family.