[Reviews · NeurIPS 2017]

Reviewer 1



The proposed submission deals with an interesting and important problem: how to automatically learn the potentially complex temporal influence structures for the multivariate Hawkes process. The proposed neutrally self-modulating multivariate point process model can capture a range of superadditive, subadditive, or even subtractive influence structures from the historical events on the future event, and the model is quite flexible. Also, the model in evaluated on both the synthetic and the real data, and yields a competitive likelihood and prediction accuracy under missing data. Comments: 1. Compared with existing work, one potential contribution of this submission is in the increased flexibility of the proposed model. First, in modeling the intensity function, a non-linear transfer function is introduced and is applied to the original defined intensity for multivariate Hawkes processes. Second, it models the hidden state h(t) as a continuous-time LSTM. 2. Since the model is more flexible and has more parameters to estimate, it will be harder to accurately learn the parameters. The algorithm basically follows the previous Back Propagation Through Time (Du et al, 2016) strategy and proposes using the Monte Carlo trick in evaluating the integral quantify in the likelihood. I think the convergence of the current algorithm might be very slow since they have too many parameters. Since the learning issue in this model is challenging, it is worthwhile discussing more on the computational efficiency. A more tractable and efficient learning algorithm might be needed in dealing with large-scale problems. 3. For the numerical section, only the likelihood under missing data is evaluated. It will be more convincing to add the results for the intensity accuracy evaluation.

Reviewer 2



The authors propose the neural Hawkes process whose intensity function is based on a continuous-time LSTM. The values of LSTM cells decay exponentially until being updated when a new event occurs, which is different from previous work of combining Hawkes process models with RNNs. The authors argue that their model has better expressivity and achieves better performance compared to the vanilla Hawkes process and the Hawkes process without positivity constraints. The overall novelty of the proposed method is not very significant, but it is well motivated and thoroughly analyzed. Some comments: (1) Why would you prefer an additive intensity function rather than multiplicative (line 229)? I assume that you always want a strictly positive intensity function and therefore choose to use the "softplus" function as your transfer function, but it seems somehow unnatural. (2) For your experiment on simulated data (Appendix C.4), it's good to see your model achieves much better performance on data generated by your model, and performs comparably or slightly better on data generated by other two models. I am interested to know whether you directly measured how well your model learns the intensity function, instead of the likelihood on held-out dataset? I suppose for your synthetic data, it's not hard to compare to the ground-truth intensity function.